

# Brief communication: An innovation-based estimation method for model error covariance in Kalman filters

Eviatar Bach[1] and Michael Ghil[1,2]

[1]Geosciences Department and Laboratoire de Météorologie Dynamique (CNRS and IPSL), École Normale Supérieure and PSL University, Paris, France
[2]Department of Atmospheric and Oceanic Science, University of California at Los Angeles, Los Angeles, United States

**Correspondence:** Eviatar Bach (eviatarbach@protonmail.com)

**Abstract.** We present a simple innovation-based model error covariance estimation method for Kalman filters. The method is based on Berry and Sauer (2013) and the simplification results from assuming known observation error covariance. We carry out experiments with a prescribed model error covariance using a Lorenz (1996) model and ensemble Kalman filter. The prescribed error covariance matrix is recovered with high accuracy.

## 1 Introduction

The Kalman filter is a state estimation method for combining model forecasts with noisy observations, and forms the basis for many data assimilation (DA) methods (Ghil et al., 1981; Kalnay, 2002). The estimation and incorporation of model error is an important aspect of the filtering problem. We briefly introduce the problem.

Following the notation of Ide et al. (1997), we assume that the true state evolution is given by

$$\mathbf{x}^{\mathrm{t}}(t_i) = \mathcal{M}_{i-1}(\mathbf{x}^{\mathrm{t}}(t_{i-1})) + \boldsymbol{\eta}(t_{i-1}), \tag{1}$$

where $\mathbf{x}^{\mathrm{t}}(t_i)$ is the true state at time $t_i$, $\mathcal{M}_i$ is the dynamic model at time $t_i$, and $\boldsymbol{\eta}$ is a model error with mean $\mathbf{0}$ and covariance $\mathbf{Q}$. The observations are given by

$$\mathbf{y}(t_i) = \mathcal{H}_i(\mathbf{x}^{\mathrm{t}}(t_i)) + \boldsymbol{\epsilon}(t_i), \tag{2}$$

where $\mathcal{H}_i$ is called the observation operator, and $\boldsymbol{\epsilon}$ is an observation error with mean $\mathbf{0}$ and covariance $\mathbf{R}$.

The standard Kalman filter assumes $\mathcal{M}_i$ and $\mathcal{H}_i$ to be linear. The extended Kalman filter (EKF) instead uses time-dependent linearizations $\mathbf{M}_i$ and $\mathbf{H}_i$, often called linear tangent models, in the estimation of the covariances, while the fully nonlinear operators are used in advancing the state $\mathbf{x}(t_i)$ itself. Ensemble Kalman filters (EnKFs) directly use the nonlinear forward model, and some EnKFs also allow for the use of nonlinear observation operators. In the rest of the paper we assume that the observation operator $\mathbf{H}$ is linear and time-independent; if it is not, a linearization can be substituted.

The Kalman filter then combines the observations $\mathbf{y}$ with model forecasts $\mathbf{x}^{\mathrm{f}}$ with covariance $\mathbf{P}^{\mathrm{f}}$, resulting in the analysis state $\mathbf{x}^{\mathrm{a}}$ with covariance $\mathbf{P}^{\mathrm{a}}$. The next forecast is then given by

$$\mathbf{x}^{\mathrm{f}}(t_{i+1}) = \mathcal{M}_i(\mathbf{x}^{\mathrm{a}}(t_i)). \tag{3}$$




The forecast error covariance $\mathbf{P}^{\mathrm{f}}$ at time $t_{i+1}$ can be estimated by

$$\mathbf{P}^{\mathrm{f}}(t_{i+1}) = \mathbf{M}_i \mathbf{P}^{\mathrm{a}}(t_i) \mathbf{M}_i^T + \mathbf{Q}(t_i), \tag{4a}$$

$$= \mathbf{P}^{\mathrm{p}}(t_{i+1}) + \mathbf{Q}(t_i), \tag{4b}$$

c.f. Ghil and Malanotte-Rizzoli (1991) or Tandeo et al. (2020); this is exact for a linear model. The first term, $\mathbf{P}^{\mathrm{p}}$, in the above estimate can be identified as the one-step predictability error (Berry and Sauer, 2013). This error is due to the effect of the system's dynamics on the initial conditions. Kalman filters, without further modification, generally only use this term, and are thus prone to underestimate $\mathbf{P}^{\mathrm{f}}$. This has led to a variety of methods for estimating $\mathbf{Q}$, often simultaneously with estimating $\mathbf{R}$; see the reviews by Duník et al. (2017) and Tandeo et al. (2020).

Then, adding the estimated $\mathbf{Q}$ to $\mathbf{P}^{\mathrm{p}}$ is known in the DA literature as additive inflation—as opposed to multiplicative inflation, where $\mathbf{P}^{\mathrm{p}}$ is multiplied by a scalar greater than 1. Additive inflation generally works better than multiplicative inflation in accounting for model errors, since multiplicative inflation assumes that model errors will span the same subspace as the errors due to initial conditions, which is not generally the case (Hamill and Whitaker, 2005).

Here, we suggest a method for estimating $\mathbf{Q}$ that is closely related to the one of Berry and Sauer (2013), but we assume that the observation noise covariance $\mathbf{R}$ is known. This assumption allows us to derive a simple estimate for $\mathbf{Q}$ that does not require either lagged innovations or the gain matrix. Nor is model linearization required in the case of an ensemble Kalman filter applied to a nonlinear forward model.

## 2    Method

Many methods for estimating $\mathbf{Q}$ rely on the statistics of the innovations $\mathbf{d}(t_i) = \mathbf{y}(t_i) - \mathbf{H}\mathbf{x}^{\mathrm{f}}(t_i)$, which equal the difference between observations and forecasts. A standard result for the Kalman filter states that

$$\mathbb{E}[\mathbf{d}(t_i)\mathbf{d}(t_i)^T] = \mathbf{H}\mathbf{P}^{\mathrm{f}}(t_i)\mathbf{H}^T + \mathbf{R}; \tag{5}$$

see, for instance, Desroziers et al. (2005) or Simon (2006, Sec. 10.1).

If the state is not fully observed, as is usually the case in DA problems, then $\mathbf{H}$ is not invertible. However, for idealized cases when $\mathbf{H}$ is invertible, we can obtain an estimate $\hat{\mathbf{Q}}$ of $\mathbf{Q}$ by substituting Eq. (4b) into Eq. (5) and rearranging:

$$\hat{\mathbf{Q}}(t_{i-1}) = \mathbf{H}^{-1}(\mathbb{E}[\mathbf{d}(t_i)\mathbf{d}(t_i)^T] - \mathbf{R} - \mathbf{H}\mathbf{P}^{\mathrm{p}}(t_i)\mathbf{H}^T)\mathbf{H}^{-T}. \tag{6}$$

See section 2.2 below for the general case in which $\mathbf{H}$ is not invertible.

In order to avoid abrupt changes in $\hat{\mathbf{Q}}$ over time, and to preserve positive semidefiniteness (see below), a temporal smoothing needs to be applied:

$$\widetilde{\mathbf{Q}}(t_{i+1}) = \rho\hat{\mathbf{Q}}(t_i) + (1 - \rho)\widetilde{\mathbf{Q}}(t_i), \tag{7}$$

where $0 < \rho < 1$ is a tunable parameter (Berry and Sauer, 2013; Tandeo et al., 2020), and $\widetilde{\mathbf{Q}}$ is the smoothed estimate. Then, $\mathbf{P}^{\mathrm{f}}(t_{i+1})$ is estimated by adding $\widetilde{\mathbf{Q}}(t_i)$ to the $\mathbf{P}^{\mathrm{p}}$ estimated by the filter. In what follows, we drop the time indices for simplicity.





Covariance matrices must be positive semidefinite: in other words, their smallest eigenvalue must satisfy $\lambda_{\min} \geq 0$. Due to the observation noise entering the $\mathbb{E}[\mathbf{dd}^T]$ term in Eq. (6), the estimate $\widetilde{\mathbf{Q}}$ can often lack this property. To avoid this problem,

a small enough $\rho$ must be chosen, and the "initial guess" $\widetilde{\mathbf{Q}}(t_0)$ should be positive semidefinite.

In general, the larger the observation noise relative to the model error, the smaller $\rho$ must be. However, if the estimated $\hat{\mathbf{Q}}$ does become indefinite at some $t_j$, definiteness can be restored. The matrix satisfying $\lambda_{\min} \geq \delta$ that is nearest in the Frobenius norm $\|\cdot\|_F$ (Horn and Johnson, 2013) to the problematic one at $t = t_j$ can be computed by using the spectral decomposition and setting all $\lambda_i < \delta$ to $\delta$ (Cheng and Higham, 1998).

## 60 2.1 Ensemble filters

In the case of an ensemble Kalman filter, we estimate $\mathbb{E}[\mathbf{dd}^T] \simeq (\mathbf{y} - \mathbf{H}\bar{\mathbf{x}}^{\mathrm{f}})(\mathbf{y} - \mathbf{H}\bar{\mathbf{x}}^{\mathrm{f}})^T$, where $\bar{\mathbf{x}}^{\mathrm{f}}$ is the mean of the forecast ensemble.

In ensemble filters, $\mathbf{P}^{\mathrm{p}}$ is estimated as

$$\mathbf{P}^{\mathrm{p}} = \frac{1}{m-1} \sum_{i=1}^{m} (\mathbf{x}_i^{\mathrm{f}} - \bar{\mathbf{x}}^{\mathrm{f}})(\mathbf{x}_i^{\mathrm{f}} - \bar{\mathbf{x}}^{\mathrm{f}})^T, \tag{8}$$

where $\mathbf{x}_i^{\mathrm{f}}$ is the $i$th ensemble member and $m$ is the ensemble size. We use this $\mathbf{P}^{\mathrm{p}}$ directly in Eq. (6), thus avoiding the need for a tangent linear model, as in Eq. (4a), when $\mathcal{M}$ is nonlinear.

Furthermore, instead of adding $\widetilde{\mathbf{Q}}$ to $\mathbf{P}^{\mathrm{p}}$, samples drawn from $\mathcal{N}(\mathbf{0}, \widetilde{\mathbf{Q}})$ can be added to each ensemble member. Mitchell and Carrassi (2015) found that this stochastic method performed better than directly adding $\mathbf{Q}$ to $\mathbf{P}^{\mathrm{p}}$, although modified deterministic methods can work better for square-root filters (Raanes et al., 2015).

## 70 2.2 Rank-deficient observations

When $\mathbf{H}$ is not invertible, we can find a solution that minimizes the Frobenius norm, as in Berry and Sauer (2013). We let $\hat{\mathbf{Q}}$ in Eq. (7) be a linear combination of fixed matrices, $\hat{\mathbf{Q}} = \sum_p q_p \mathbf{Q}_p$. This formulation can be used to specify a simplified structure, such as a diagonal matrix or a block-constant one.

Let $\mathbf{q}$ be the vector of coefficients $\{q_p\}$. Then,

$$75 \quad \mathbf{q} = \underset{\{q_p\}}{\arg\min} \left\| \mathbf{C} - \sum_p q_p \mathbf{H}\mathbf{Q}_p \mathbf{H}^T \right\|_F, \tag{9}$$

where

$$\mathbf{C} = \mathbb{E}[\mathbf{dd}^T] - \mathbf{R} - \mathbf{H}\mathbf{P}^{\mathrm{p}}\mathbf{H}^T. \tag{10}$$

The minimization in Eq. (9) is carried out by finding the least-squares solution of

$$\mathbf{A}\mathbf{q} \simeq \mathrm{vec}(\mathbf{C}), \tag{11}$$

where the $p$th column of $\mathbf{A}$ is $\mathrm{vec}(\mathbf{H}\mathbf{Q}_p\mathbf{H}^T)$.



## 3 Numerical experiments

We apply the proposed method to the Lorenz (1996) model:

$$\frac{\mathrm{d}x_i}{\mathrm{d}t} = -x_{i-2}x_{i-1} + x_{i-1}x_{i+1} - x_i + F, \tag{12}$$

where we use 40 variables, the indices are cyclical, and $F = 8$. The characteristic time of the model—measured by the reciprocal of the largest Lyapunov exponent—is about 0.6, and we apply a 4th-order Runge–Kutta scheme with $\Delta t = 0.05$ to integrate it.

The true $\mathbf{Q}$ matrices are prescribed as follows. We generate a $40 \times 40$ band matrix $\mathbf{B}$ with bandwidth 20, and the numbers on the band are drawn from a uniform distribution $\mathcal{U}(0,1)$. The experiments are carried out with two different orders of magnitude of the model error, one with $\mathbf{Q}_1 = (1/10)(\mathbf{B} - 0.4\mathbf{J}_{40})(\mathbf{B} - 0.4\mathbf{J}_{40})^T$ and the other with $\mathbf{Q}_2 = (1/100)(\mathbf{B} - 0.4\mathbf{J}_{40})(\mathbf{B} - 0.4\mathbf{J}_{40})^T$, where $\mathbf{J}_{40}$ is the $40 \times 40$ matrix of ones. The matrix $\mathbf{Q}_1$ is shown in Fig. 2a.

We carry out three distinct experiments:

1. Model error is $\mathbf{Q}_1$ and the state is fully observed.

2. Model error is $\mathbf{Q}_1$ but only every second $x_i$ is observed. In this experiment, furthermore, we parameterize $\mathbf{Q}$ as a block-constant matrix of $4 \times 4$ blocks.

3. Model error is $\mathbf{Q}_2$ and the state is fully observed.

For each experiment, we use the ensemble transform Kalman filter (ETKF: Bishop et al., 2001) with 80 ensemble members. At every timestep, we draw a vector from the multivariate normal distribution $\mathcal{N}(\mathbf{0}, \mathbf{Q})$ and add it to all the ensemble members. We take $\mathbf{R} = 0.4\mathbf{I}$, with $\rho = 10^{-3}$ for experiment 1 and $\rho = 10^{-4}$ for experiments 2 and 3. The estimate $\widetilde{\mathbf{Q}}$ is initialized with $0.1\mathbf{I}_{40}$ for experiments 1 and 3 and with $\mathbf{I}_{40}$ for experiment 2. The number of DA cycles is 3 000 for experiment 1, 20 000 for experiment 2, and 15 000 for experiment 3.

### 3.1 Results

Figures 1 and 2b shows the results of experiment 1. After about 2 000 DA cycles, the error in estimating $\mathbf{Q}_1$ stabilizes (Fig. 1a), which also results in lower analysis errors when $\widetilde{\mathbf{Q}}_1$ is used in the DA process (Fig. 1b). Furthermore, $\mathbf{Q}_1$ is recovered with high fidelity (Fig. 2b). Note that, for the analysis errors, we use the continuous ranked probability score (CRPS: Hersbach, 2000), a probabilistic error metric, to measure the discrepancy between the ensemble and the true probability distributions.

Figure 2c shows the estimate of $\mathbf{Q}_1$ obtained in experiment 2, with partial observations and a block-constant formulation. In this case, too, the method successfully recovers the coarse structure of $\mathbf{Q}_1$.

Finally, Figure 3 shows the results of experiment 2. Here, since $\rho$ is smaller, it takes longer for the error to stabilize. The structure $\mathbf{Q}_2$ is also recovered with high fidelity (not shown).





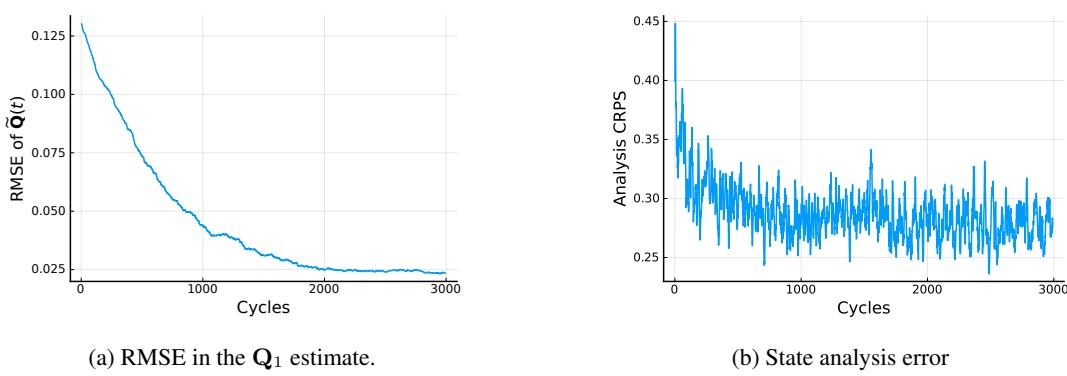

(a) RMSE in the $\mathbf{Q}_1$ estimate.

(b) State analysis error

**Figure 1.** Numerical results for experiment 1: (a) the root-mean-square error (RMSE) in the $\mathbf{Q}_1$ estimate; and (b) analysis error in the state $\mathbf{x}$, measured by the continuous ranked probability score (CRPS). A 10-timestep moving average is applied in (b).

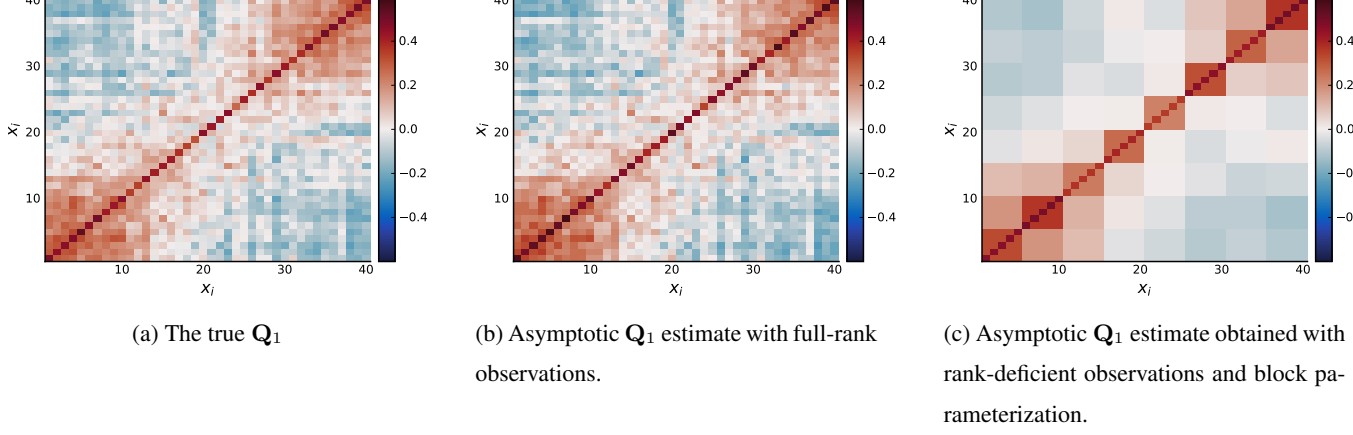

(a) The true $\mathbf{Q}_1$

(b) Asymptotic $\mathbf{Q}_1$ estimate with full-rank observations.

(c) Asymptotic $\mathbf{Q}_1$ estimate obtained with rank-deficient observations and block parameterization.

**Figure 2.** Asymptotic estimates of the model error covariance $\mathbf{Q}_1$: (a) true $\mathbf{Q}_1$; (b) results of experiment 1; and (c) results of experiment 2.

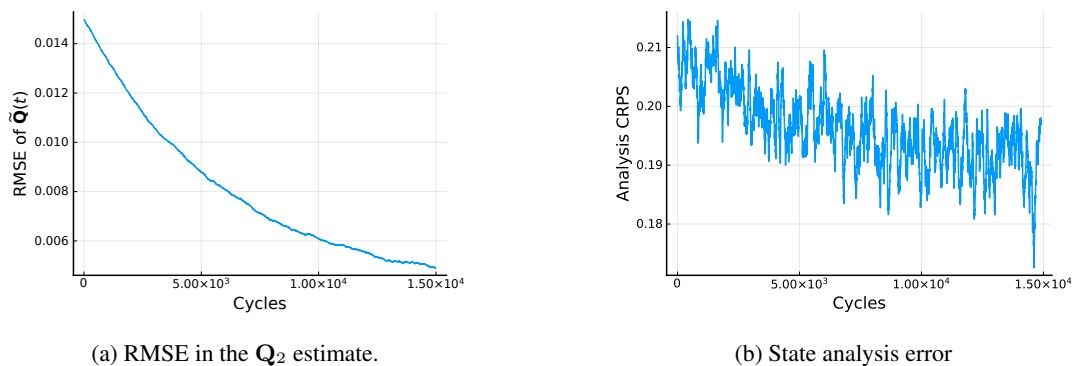

(a) RMSE in the $\mathbf{Q}_2$ estimate.

(b) State analysis error

**Figure 3.** Same as Fig. 1, but for experiment 3. A 100-timestep moving average is applied in (b).



## 4   Conclusions

We present a simple method for estimating the model error covariance matrix in a Kalman filter. When applying an ensemble Kalman filter, our method does not require lagged innovations, nor the gain matrix or the linear tangent model. The estimated model error covariance can then be added to the forecast covariance estimated by the filter. Such a form of additive inflation generally performs better than the multiplicative inflation more often used in the literature.

*Code availability.* The Julia code implementing this method is available at https://github.com/eviatarbach/model_error_estimation.git

*Author contributions.* E.B. devised the method, carried out the computations and drafted the paper. M.G. contributed to the analysis of the results and to the final version of the paper.

*Competing interests.* No competing interests are present.

*Acknowledgements.* E.B. was funded by the Make Our Planet Great Again (MOPGA) postdoctoral program.





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
