# Peer review of "Brief communication: An innovation-based estimation method for model error covariance in Kalman filters"

_Nonlinear Processes in Geophysics, 2021_

## Referee Comment (RC1)

Review comments for brief communication **npg-2021-35**

*An innovation-based estimation method for model error covariance in Kalman filters*

December 8, 2021

This manuscript is concerned with the estimation of the model error covariance matrix, $\mathbf{Q}$, in Kalman filtering and related methods. A method is proposed that uses the discrepancy between the expected and observed innovations, averaged in time.

Regrettably, I struggle to find anything new and noteworthy in this manuscript, and wonder if the authors have made any effort at surveying the existing literature. For example, 'Adaptive Kalman filtering' usually gets a chapter or a section in textbooks [Jazwinski, 1970; Anderson and Moore, 1979], while the estimation of system matrices is already a mature field in data assimilation [Mitchell and Houtekamer, 2000]. In this context, a special case of the method of Berry and Sauer [2013], which is already using the most basic estimator, is not interesting.

**Other issues**

- The hyperlink to the code is dead.
- $\hat{\mathbf{Q}}$ appears to be defined via the expectation, $\mathbb{E}$, both in equation (6) and on line 54.
- Please be more specific on line 97. What is $\mathbf{Q}$ here? Isn't it unknown? Do you only draw a single vector?
- Comment on need for such a small $\rho$ in your experiments.
- Please motivate the design of $\mathbf{Q}_1$ and $\mathbf{Q}_2$.
    - Why is *banded* $\mathbf{B}$ needed? Note that any $\boldsymbol{A}\boldsymbol{A}^t$ would form a covariance matrix.
    - Why is $\mathbf{J}_{40}$ being subtracted.
- Include the upper and lower tick on the colorbars.

**References**

B. D. O. Anderson and J. B. Moore. *Optimal Filtering.* Prentice-Hall, Englewood Cliffs, NJ, 1979.

Tyrus Berry and Timothy Sauer. Adaptive ensemble Kalman filtering of non-linear systems. *Tellus A*, 65, 2013.

A. H. Jazwinski. *Stochastic Processes and Filtering Theory*, volume 63. Academic Press, 1970.

Herschel L. Mitchell and P. L. Houtekamer. An adaptive ensemble Kalman filter. *Monthly Weather Review*, 128 (2):416–433, 2000.

---

## Referee Comment (RC2)

**"Brief communication: An innovation-based estimation method for model error covariance in Kalman filters"**

Revision for NPG, received on November 15, 2021.

The article discusses a method for estimating the Q error covariance matrix of a dynamic model in ensemble data assimilation. This matrix is important because it plays the role of additive inflation in EnKF. The proposed methodology assumes that the error covariance matrix R of the observations is known, which is a strong assumption. Due to this simplification, Q is estimated using the second-order moment of the innovation. The proposed method is online and therefore dependent on two adjustment parameters, a forgetting factor and a first guess on Q. These parameters are important in practice. The article needs to take into account the estimation of these tuning parameters.

**Major comments:**
- The proposed methodology is an online estimation method, meaning that Q(t) is estimated synchronously with the state x(t). Authors suggest that Q is contact in time but they could have considered a time varying Q matrix.
- The proposed methodology is highly dependent on \rho, the forgetting factor, and Q(t_0), the initial model error covariance. The estimation of an adaptive \rho parameter should be addressed in this paper. Moreover, iterative procedures like the EM algorithm (Dreano et al. 2017 or Pulido et al. 2018) should be considered to estimate Q(t_0).

**Minor comments:**
- L. 52, please write P^p(t_{i+1}).
- L. 53-59, not sure the discussion is useful.
- L. 79, what is the meaning of "vec"?
- L. 87, not sure to understand the explanation of "bandwidth 20", can you clarify?
- L. 90, looking at the Lorenz-96 equations, x_1 and x_40 are neighbors and should be positively correlated. This is not what is shown in Fig.2 (a). I think you should consider such covariance between neighbors in Q.
- L. 98, please remind the reader that \rho is the forgetting factor.

**References:**

- Dreano, D., Tandeo, P., Pulido, M., Ait-El-Fquih, B., Chonavel, T., & Hoteit, I. (2017). Estimating model-error covariances in nonlinear state-space models using Kalman smoothing and the expectation–maximization algorithm. Quarterly Journal of the Royal Meteorological Society, 143(705), 1877-1885.
- Pulido, M., Tandeo, P., Bocquet, M., Carrassi, A., & Lucini, M. (2018). Stochastic parameterization identification using ensemble Kalman filtering combined with maximum likelihood methods. Tellus A: Dynamic Meteorology and Oceanography, 70(1), 1-17.